

# COMICS: a community property-based triangle motif clustering scheme

Yufan Feng[1], Shuo Yu[1], Kaiyuan Zhang[1], Xiangli Li[1] and Zhaolong Ning[1,2]

[1] School of Software, Dalian University of Technology, Dalian, China
[2] State Key Laboratory for Novel Software Technology, Nanjing University, Nanjing, China

## ABSTRACT

With the development of science and technology, network scales of various fields have experienced an amazing growth. Networks in the fields of biology, economics and society contain rich hidden information of human beings in the form of connectivity structures. Network analysis is generally modeled as network partition and community detection problems. In this paper, we construct a community property-based triangle motif clustering scheme (COMICS) containing a series of high efficient graph partition procedures and triangle motif-based clustering techniques. In COMICS, four network cutting *conditions* are considered based on the network connectivity. We first divide the large-scale networks into many dense subgraphs under the cutting *conditions* before leveraging triangle motifs to refine and specify the partition results. To demonstrate the superiority of our method, we implement the experiments on three large-scale networks, including two co-authorship networks (the American Physical Society (APS) and the Microsoft Academic Graph (MAG)), and two social networks (Facebook and gemsec-Deezer networks). We then use two clustering metrics, compactness and separation, to illustrate the accuracy and runtime of clustering results. A case study is further carried out on APS and MAG data sets, in which we construct a connection between network structures and statistical data with triangle motifs. Results show that our method outperforms others in both runtime and accuracy, and the triangle motif structures can bridge network structures and statistical data in the academic collaboration area.

## INTRODUCTION

In all aspects of human endeavor, we are in the world of large-scale data, embracing the aspects of biology, medicine, social, traffic, and science (*Ning et al., 2017*). These data sets describe the complicated real-world systems from various and complementary viewpoints. Generally, the entities in real-world systems are modeled as nodes, whose connections and relationships are modeled as edges. Those networks become new carriers of rich information from domain-specific areas, such as the reciprocity among people in online social networks (*Koll, Li & Fu, 2013*). More than that, human beings are inclined to cooperate or participate in group activities, which can be reflected in social and academic collaboration networks. To be more specific, in academic area, big scholarly data grows rapidly, containing millions of authors, papers, citations, figures, tables, and other

Corresponding author
Zhaolong Ning,
zhaolongning@dlut.edu.cn

massive scale related data, such as digital libraries and scholarly networks (*Xia et al., 2017*). As collaboration behaviors among scholars are becoming frequent, collaboration networks are generally in large-scale and contain rich collaboration information, reflecting the cooperation patterns among scholars in different research areas. *Bordons et al. (1996)* regard the academic teams as scientists communities, in which scholars can share research methods, materials, and financial resources rather than institutions organized by fixed structures (*Barjak & Robinson, 2008*). Furthermore, the ternary closures in social networks constitute a minimal stable structure; that is, a loop with three nodes. The number of ternary closures in social networks changes over time, which reveals the evolvement of human social behaviors. Besides, the definition of a clustering coefficient is based on the distributions of ternary closures. *Milo et al. (2002)* defined small network structures as motifs to present interconnections in complex networks by numbers that are significantly higher than those in randomized networks. Motifs can define universal classes of networks, and researchers are carrying on the motif detection experiments on networks from different areas, such as biochemistry, neurobiology, and engineering, to uncover the existence of motifs and the corresponding structure information in networks (*Ribeiro, Silva & Kaiser, 2009*; *Bian & Zhang, 2016*). Hence, triangle motifs can be used to uncover relationships in networks.

Connectivity is a fundamental character in both graph theory and network science. When networks are in small-scale, the dense areas can be easily identified. However, with the rapid growth of network scale and diversity, many graph partition methods, community detection, and clustering algorithms fail to uncover the information of graph structure. Graph partition and mining algorithms consume a large amount of time when dealing with large-scale networks, for example, the gSpan algorithm (*Yan & Han, 2002*) and the Min–Cut algorithm (*Stoer & Wagner, 1997*), which overlook the elementary network structures. The clusters and subgraphs of a large network are generally have small internal distances and large external distances among nodes. Considering the ternary closures, triangle network motifs have been regarded as elementary units in networks. However, a general method to cluster the communities and analyze the relationships with community properties and triangle motifs effectively is still lacking.

In this paper, we propose a community property-based triangle motif clustering scheme (COMICS) to cluster network communities, and analyze the relationships with triangle motifs. In this method, we partition networks with the edge connection properties and regard the undirected and unweighted complete triangle motifs as the element clustering units. The partition operations are based on four network cutting *conditions*, whose definitions are based on the network connectivity to maintain the massive links in networks. More than that, by considering the American Physical Society (APS) and Microsoft Academic Graph (MAG) data sets in the academic analysis area, we regard each cluster generated from the input network as an academic team, and define three metrics: teamwork of collaborator variance (TCV), teamwork of paper variance (TPV), and motif variances of scholars (MSV) to evaluate the

behaviors of the detected academic teams. Our contributions can be summarized as follows:

- By jointly considering time complexity and clustering accuracy, we construct the COMICS, which mines the structure information with complete triangle motifs. A series of speed-up and refining methods, graph partition and refining techniques, are integrated to improve the performance of the basic clustering process.
- We prove the time complexity of the presented algorithm is $O(rn^3)$, where $r$ is the number of the clustered subgraphs from the original large network, and $n$ is the number of nodes.
- We regard the undirected and unweight complete triangle motif as the elementary unit instead of nodes in the clustering procedure. Our work verifies that the complete triangle motif is available in network analysis.
- We define three metrics to analyze the hidden information in academic collaboration networks. Performance evaluations show that the academic teams with high quantity of scholar motif variances also have high values of TCVs.

The roadmap of this paper is illustrated as follows. We briefly illustrate the related works in the following section. After that, a series of fundamental definitions, problem statement, and some necessary notations are described. Then, we describe the architecture of COMICS in details. We evaluate the performance of our method with three large-scale networks as case studies in the experiment section. Finally, we conclude this paper.

## RELATED WORK

Information mining from large-scale networks is a significant research topic, reflecting the connecting patterns, and the social relationships among different entities (*Shi et al., 2017*; *Schaeffer, 2007*). In collaboration networks, the information reflects the collaboration patterns and academic social relationships among scholars in different disciplines. Community detection is traditionally considered as a kind of graph partition to discover exhaustive and disjoined node clusters in a given network (*Khan et al., 2017*). The discovery of structures in networks has attracted scholars' attentions for a long time. The authors in *Leskovec et al. (2009)* explore from a novel perspective to identify meaningful communities in large social and information networks. They notice that large networks have very different structures. For example, different transcription networks from *Escherichia coli* and *Saccharomyces cerevisiae* have large differences in the frequency motif structures (*Wegner, 2014*). Over the past years, a number of graph clustering methods have been investigated. For example, evolutionary algorithms (EAs) have been proposed and applied successfully in the network optimization and clustering problems (*Gong et al., 2012*). Recently, scholars have successfully developed both single- and multi-objective EAs to discover internal structure information of networks (*Li & Liu, 2016*; *Pizzuti, 2012*). A particle swarm optimization algorithm is put forward, which reveals community structures in large-scale social networks (*Cai et al., 2015*). In *Girvan & Newman (2002)*, node centrality and betweeness centrality were used to extract

communities in a network. Since modularity is becoming very popular by partitioning networks into non-overlapping subgraphs, modularity score, compactness-isolation, and other criteria are leveraged to evaluate functions in large graph partition problems *Bagrow (2008)*.

A number of large-scale partition methods based on community detection start with a given seed, and then expand by iteratively adding the neighboring node that contributes most to the score function, until the score function stops improving (*Luo, Wang & Promislow, 2008*; *Ma et al., 2014*). The authors in *Du et al. (2007)* developed an efficient community detection method in social networks, which combines the topological information with the entity attributions to detect network communities. However, this method works for merely part of network structures.

Motifs in networks are small connected subnetworks, occurring in significant high frequencies and have recently gathered attentions. Motifs in networks have been studied as elementary structures in complex network analysis (*Shervashidze et al., 2009*). In hypergraphs, the clustering algorithms mainly focus on transforming the hypergraphs into simple graphs (*Zhou, Huang & Schölkopf, 2006*). Then, the simple graphs can be clustered with spectral clustering procedures based on the normalized Laplacian matrices (*Li & Milenkovic, 2017*). In that case, the motifs can be constructed with nodes from different graph layers of hypergraphs (*Zhou, Huang & Schölkopf, 2006*), for example, a triangle motif can be used to represent one heterogeneous hypergraph with three different layers. More than that, conductance is a vital definition in spectral clustering (*Louis, 2015*; *Li & Milenkovic, 2018*). Hence, in large-scale networks, spectral clustering and motif are combined to large-scale network clustering. Triangle motif structures guarantee the structural connections. The motif-based conductance ensures the applicability spectral clustering in large-scale networks. Some local graph clustering methods have been investigated by incorporating high-order network information captured by small subgraphs (*Yin et al., 2017*; *Lee, Gharan & Trevisan, 2014*; *Li et al., 2017b*). In *Wegner (2014)*, the authors define a subgraph cover to represent the network with motifs. The cover consists of a set of motifs and their corresponding frequencies. Besides, the network motifs can be detected by comparing the frequencies of subgraphs in the original network with a statistical model. The authors in *Wegner (2014)* notice that real networks contain significant densities of different motifs. It illustrates networks in different fields hold different collaboration patterns, and motifs are the fingerprints of different networks. By observing the characteristics from real networks, *Benson, Gleich & Leskovec (2016)* develop a generalized framework to cluster networks on basis of higher-order connected patterns. A framework is proposed to model the relations between higher-order motif instances and graph nodes with a bipartite graph (*Li et al., 2017a*). In *Monti, Otness & Bronstein (2018)*, MotifNet is introduced to deal with directed graphs by exploiting local graph motifs. In order to tackle the graph analysis problem, we combine the graph partition method with the motif-based clustering procedure to speed up the clustering process.

## System model and problem formulation

In this section, we present basic theoretical definitions about cutting *conditions* and the mathematical expressions of motifs. After that, we describe the investigated problems in details.

## Comparative *conditions* for cluster

In this subsection, we introduce four conditions to partition the original large collaboration networks into different clusters.

For a given graph $G = (V, E)$, we define the adjacent matrix $H = \{h_{i,j}\}$ as: if there exists an edge between vertices $i$ and $j$, $h_{i,j} = 1$; otherwise, $h_{i,j} = 0$.

Network partition is defined as: $\mathscr{P} = \{G_1, G_2, \ldots, G_k\}$, $1 \leq k \leq |V|$, subject to: (1) $\bigcup_{k=1}^{K} G_k = V$, (2) $G_k \cap G_t = \varnothing, \forall k \neq t$, and (3) $G_k \neq \varnothing, \forall k$. For $\forall k$, the partition $\mathscr{P}$ satisfies the $x-valid$ condition, called $x-valid$ cluster partition of $G$, and $x$ is defined as the following conditions according to *Lu et al. (2013)*:

$$\text{Condition 1}: \sum_{\forall j \in G_k} H_{i,j} > \sum_{\forall j \in V \setminus G_k} H_{i,j}, \ \forall i \in G_k, \ \forall k. \tag{1}$$

$$\text{Condition 2}: \sum_{\forall j \in G_k} H_{i,j} > \sum_{\forall j \in G_t} H_{i,j}, \ \forall i \in G_k, \ k \neq t. \tag{2}$$

$$\text{Condition 3}: \sum_{\forall j \in G_k} H_{i,j} > \sum_{\forall i \in G_k, \forall j \in V \setminus G_k} H_{i,j}, \ \forall k. \tag{3}$$

$$\text{Condition 4}: \sum_{\forall j \in G_k} H_{i,j} > \sum_{\forall i \in G_k, \forall j \in G_t} H_{i,j}, \ \forall k \neq t. \tag{4}$$

Conditions 1 and 2 check the validity of clusters at the vertex level to confirm whether the internal degree of each vertex is larger than that of the external degree. Conditions 3 and 4 check the validity of clusters, that is, comparing the total internal degree of each cluster. When large graphs are partitioned under the above-mentioned four conditions, Condition 1 generally results in fewer, but larger subgraphs; Condition 4 will lead to more and smaller communities; Conditions 2 and 3 will cause more and smaller communities than Condition 1, but fewer and larger communities than Condition 4.

## Definitions of network triangle motifs

In real networks, the most common high-order structures are small network subgraphs, which are defined as motifs, that is, a set of edges with a small number of nodes. In this paper, we analyze undirected triangular motif-based networks.

Formally, we define a triangle motif by a tuple $(B, A)$, where $B$ is a $3 \times 3$ binary matrix and $A \subset \{1, 2, \cdots, n\}$ is the set of anchor nodes. The matrix $B$ encodes the edge pattern between the three nodes in triangle motifs, and $A$ represents a relevant subset of nodes to define motif conductance. Then, let $\chi_A$ be a selection function, taking the subset of a 3-tuple induced by $A$. Define set $(\cdot)$ as the operator, which takes a tuple to a set, set $(v_1, v_2, v_3) = \{v_1, v_2, v_3\}$. Then, the motif set of an unweighted and undirected graph with adjacency matrix $A$ can be denoted by Eq. (5), where $v_1 \neq v_2 \neq v_3$,

$$B_1 = \begin{pmatrix} 0 & 1 & 1 \\ 1 & 0 & 1 \\ 1 & 1 & 0 \end{pmatrix} \quad A_1 = \{1, 2, 3\} \quad M_1(B_1, A_1) = \{(\{a, b, d\}, \{a, b, d\}), (\{a, b, e\}, \{a, b, e\})\}$$

$$B_2 = \begin{pmatrix} 0 & 1 & 1 \\ 1 & 0 & 0 \\ 1 & 0 & 0 \end{pmatrix} \quad A_2 = \{2, 3\} \quad M_2(B_2, A_2) = \left\{ \begin{matrix} (\{a, c, d\}, \{c, d\}), (\{a, d, e\}, \{d, e\}), \\ (\{b, d, e\}, \{d, e\}) \end{matrix} \right\}$$

**Figure 1** **Example of motif definition in diagram: the motifs Tri$_1$ and Tri$_2$ are leveraged to detect in the five-node graph $G$ on the left figure.** The motifs are defined by a binary matrix $B$ and an anchor set of nodes. $B_1$ and $B_2$ are the binary matrices of Tri$_1$ and Tri$_2$, respectively. Similarly, $A_1$ and $A_2$ are the anchor node sets of Tri$_1$ and Tri$_2$, respectively.

$$\text{Tri}(B, A) = \{\text{set}(v), \text{set}(\chi_A(v)) | v \in V^k, A_v = B\}. \tag{5}$$

Here, $A_v$ is the $k \times k$ adjacency matrix of the subgraph with $k$ nodes of the order vector $v$. In this paper, the motifs are undirected and unweighted. The matrix $B$ of motif is symmetrical. Hence, we use $(v, \chi_A(v))$ to denote $(\text{set}(v), \text{set}(\chi_A(v)))$ for convenience. Furthermore, we regard any $(v, \chi_A(v)) \in \text{Tri}(B, A)$ as a motif instance. If $A$ and $B$ are arbitrary or clear from context, we simply denote the motif set by Tri. We define the motifs, that is, $\chi_A(v) = v$, as simple motifs, and others are anchor motifs.

We give an example of triangle motif definition, as shown in Fig. 1, aiming to cluster the given five-node network by the two triangle motifs. First, we define the motifs by the description in Eq. (5). For motif Tri$_1$, there are two instances of the motifs in $G$. Meanwhile, for motif Tri$_2$, $G$ has three instances, and the anchor sets of each instance is the node whose degree is one.

The definition of the triangle motif conductance replaces an edge with a motif instance of type Tri. We suppose that a given network has been clustered into two subnetworks, that is, $g$ and $\bar{g}$, and the conductance based on motifs can be expressed in Eq. (6),

$$\psi_{\text{Tri}}^{(G)}(S) = \frac{(cut_{\text{Tri}}^{(G)}(g, \bar{g}))}{min(vol_{\text{Tri}}^{(G)}(g), vol_{\text{Tri}}^{(G)}(\bar{g}))}. \tag{6}$$

When there is at least one anchor node in $S$ and at least one anchor node exists in $\bar{g}$, a motif instance can be cut. In Eq. (6), $cut_{\text{Tri}}^{(G)}(g, \bar{g})$ is the number of instance cut. $vol_{\text{Tri}}^{(G)}(g)$ is the number of instances, whose end nodes are in $g$. To be more specific, following the definition of Tri in Eq. (5), as for the same $\chi_A(v)$, there may exist many different values of $v$, and nodes in $\chi_A(v)$ are still counted proportionally into the number of motif instances. This growth tendency of motifs is consistent with the number of nodes in networks. This can prove the availability of motifs in clustering networks.

## Definition of motif-base matrices

Given a graph and a set of motif Tri, the motif adjacency matrix $W_{\text{Tri}}$ of graph is shown as:

$$(W_{\text{Tri}})_{ij} = \sum_{((v, \chi_A(v)) \in \text{Tri})} 1\left(\{i, j\} \subset \chi_A(v) | i \neq j\right). \tag{7}$$

Herein, $(W_{\text{Tri}})_{ij}$ is the number of motif instances in $M$, where nodes $i$ and $j$ are both in a triangle motif. In other words, the weight will be added into $(W_{\text{Tri}})_{ij}$ if and only if node $i$

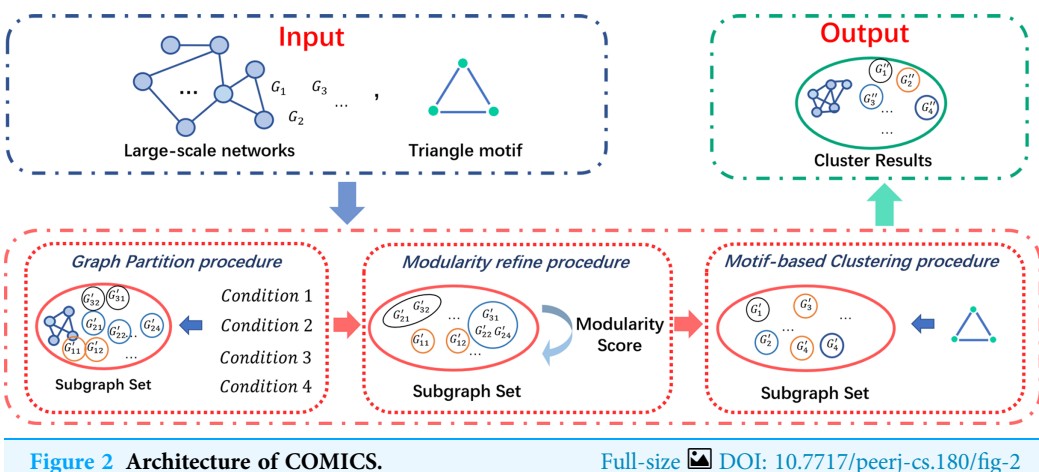

**Figure 2  Architecture of COMICS.**   

and node $j$ both appear in the anchor set. In the collaboration networks, $(W_{\text{Tri}})_{ij}$ depends on the number of scholars, who collaborate with both scholar $i$ and scholar $j$.

Then, the motif diagonal degree matrix $D_M$ is defined as $(D_{\text{Tri}})_{ii} = \sum_{j=1}^{n}(W_{\text{Tri}})_{ij}$. The motif Laplacian can be calculated by $L_{\text{Tri}} = D_{\text{Tri}} - W_{\text{Tri}}$. Finally, we normalize the motif Laplacian as:

$$\Gamma_{\text{Tri}} = I - D_{\text{Tri}}^{-1/2} W_{\text{Tri}} D_{\text{Tri}}^{-1/2}. \tag{8}$$

## Problem statement

Let $G = (V, E)$ be a connected large network, where $V$ is the node set, and $E$ is the edge set. If $G$ contains several disjoint networks, it can be expressed as $G = \{G_1, G_2, \cdots, G_n\}$. The complete triangle is the target motif to analyze the large-scale networks. Our objective is to find the dense and stable disjoined subgraph set $\mathscr{P} = \{G_1'', G_2'', \ldots, G_m''\}$ of the given network by motifs.

Given a node $v \in V$ in the network $G_i \in G$, the degree of $v$ is denoted by $\deg(v)$ and the neighbor node set of the subgraph $G_i$ in the original networks is denoted by $N_{G_i}$. In the partition phase, $G_i$ can be cutted into a set of subgraphs, $\{G_{i1}', G_{i2}', \ldots, G_{ik}'\}$. For a node $v$ in a partition subgraph $G_{ij}'$ of $G_i$, we use $\deg_{\text{inter}}(v)$ and $\deg_{\text{extra}}(v)$ to represent the degree within $G_{ij}'$ and the number of edges between $G_{ij}'$ and $G_i/G_{ij}'$, respectively. $\text{Con}(G_{ij}')$ is the set of subgraphs that are connected with $G_{ij}'$ in the partition $\mathscr{P}$. Variable $Q$ is the modularity score when the original graph $G$ gets the partition $\mathscr{P}$. In the process of graph partition, we cut the original networks into initial subgraphs under the four conditions. In that way, the network can be cutted into subgraphs with strong internal connectivity and weak external connectivity in both local and global aspects. The modularity score refining subprocedure can optimize the partition. Then, we cluster and analyze the initial dense subgraphs by the complete triangle motif.

## COMICS algorithm

In this section, we describe the whole process of our COMICS in details. As shown in Fig. 2, COMICS consists of a series of partition refine strategies and a motif-based clustering procedure, that is, graph partition, modularity refine procedure and motif-based

---

**Algorithm 1** Graph Partition Algorithm

---

**Input:** Large graph $G$, *conditions*

**Output:** $R$: A partition set $\mathscr{P}$ of $G$

1: Add $G$ to $R$

2: **while** $|R|$ increases **and** $|R| \neq 1$ **do**

3:   **for** each subgraph $G_i$ in $R$ **do**

4:     \\ $root_{G'_{ij}}$ is a node from $G_i$. A new subgraph $G'_{ij}$ can be generated from $G_i$ with $root_{G'_{ij}}$.

5:     $root_{G'_{ij}} = \underset{v \in V}{\operatorname{argmin}}\{deg(v)\}$

6:     **for** node $v$ in $N(G'_{ij})$ **do**

7:       **if** $v$ satisfies the given *conditions* **then**

8:         Add *node* to $G'_{ij}$

9:       **else**

10:        $root_{G'_{ij}} = \underset{v \in V}{\operatorname{argmin}}\{D(v)\}$

11:      **end if**

12:    **end for**

13:    Make the partition $\left(G'_{ij}/G_i, G_i\right)$

14:  **end for**

15: **end while**

16: return $R$

---

clustering procedure. We first illustrate the graph partition techniques under four conditions and the modularity refine procedure in Algorithm 1. After that, the motif-based clustering procedure is constructed on each subgraph in cutting set. We specify the whole clustering layer algorithm in Algorithm 4, by which we are able to get the close and stable subgraph structures from the original input networks.

## Graph partition and modularity refine procedures

To obtain the total information of large networks effectively, we first perform cutting operations in large networks. In this subsection, we explain the graph partition and modularity refine procedures in details. We use the total large graph as the input of the partition procedure, and the procedure returns a set of partition subgraphs of the original graph. The subgraph set is refined in the modularity refine procedure by modularity score. In the graph partition procedure, we take the differences between the internal and external degrees as the degree difference value of a node $v$, denoted by $D(v)$, that is,

$$D(v) = deg_{inter}(v) - deg_{extra}(v). \tag{9}$$

For all pairs of nodes $v$ and $u$ in networks, if nodes $v$ and $u$ fall in the same subgraph, the quantity of $s_v s_u$ is 1, otherwise, it equals to −1. $|E|$ is the total number of edges in the original network. The value of $e_{v,u}$ is 1, if there exists one edge between nodes $v$ and $u$, otherwise it is 0. Therefore, the modularity score of a network is defined as Eq. (10):

$$Q = \frac{1}{4|E|} \sum_{v,u} \left( e_{v,u} - \frac{deg(v)deg(u)}{2|E|} \right) (s_v s_u + 1). \tag{10}$$

As described in Algorithm 1, we take the large networks and the cutting conditions as the input of the graph partition procedure. At the beginning of this procedure, there is only one original graph added in the result partition set $R$. The number of subgraphs in $R$ is represented as $|R|$. In each outer loop, each subgraph $G_i$ is chosen to generate new subgraphs, $G'_{ij}$. Hence, the root node of the new subgraph $G'_{ij}$ is selected randomly among the nodes that are with the minimum degree of the new subgraph in $R$. As described in lines 6–12, the loop aims to generate the new graph from the root node from $G_i$. If at least one neighbor node of the total new subgraph satisfies the validity of the input *conditions*, the neighbor node is added to the new subgraph $G'_{ij}$ with its connectivity; otherwise, it means that there is no neighbor with this root node, and we select a new node in $G_i$ as the root to check the connectivity with the original network. The node with the minimum difference between the internal and external degrees will be the root node of $G'_{ij}$. Then, graph partition operations will be carried until no other nodes from the original networks satisfy the cutting condition.

In line 3 of Algorithm 1, there are two cases when selecting a new root node of the new subgraph: one is that the root node selected in line 4 or line 9 has no neighbor, that is, the degree of root node is 0. It indicates the root node $v_{\text{root}}$ is invalid, and no new subgraph can be added into $R$. The other situation is that there is no node in $G$ connected with the partition subgraph. In other words, the iteration of adding nodes to the new subgraph stops. Then a new subgraph generated by the root node $v_{\text{root}}$ is added to the subgraph set $R$. The iteration of the partition procedure stops when the number of subgraphs in $R$ does not increase any more. However, if there exists one graph in $R$ all the time, it illustrates that the root node with the minimum degree is invalid, and we have to choose another root node and restart the iteration.

Algorithm 1 cuts large graph into small dense subgraphs. Each loop generates a new subgraph from set $R$, and the loop stops when no more dense subgraph can be found. To avoid damaging the connectivity of the rest nodes, we check the connectivity of both cutting and remaining parts, if there exists more than one component, we put all subgraphs in set $R$ to the modularity refine procedure in Algorithm 2.

The modularity refine procedure described in Algorithm 2 takes the results of the Algorithm 1 as input to refine the partition results of the original networks. As shown in Algorithm 2, lines 1 and 2 enumerate two connected subgraphs: $G'_{ij}$ and $G'_{ik}$ in $R$. In the following line 3 to line 6, if the two subgraphs are combined to one, it results in much higher modularity score than the original network partition. Then we replace $G'_{ij}$ and $G'_{ik}$ by $G'_{ij} \cup G'_{ik}$, and the iteration stops until the modularity scores do not increase any more. Variable $R_0$ is the result set of Algorithm 2, containing a series of refined subgraphs. This procedure of graph partition aims to maintain more structure information of the original large-scare networks, so that the output partition by Algorithm 2 can achieve higher modularity score than the input subgraph set. The operation of merging these two cutting subgraphs increases the internal degrees. Merging two subgraphs into one can also decrease the external degree for the subgraphs in partition $\mathscr{P}$. If two subgraphs can be merged, the number of edges between them is larger than that cannot be

---

**Algorithm 2** Modularity Refine Algorithm

**Input:** Subgraphs set $R$, empty set $R_0$

**Output:**   Refined subgraphs set $R_0$

1: **for** $G'_{ij}$ *in* $R$ **do**

2:   **for** $G'_{ik}$ in $\mathrm{Con}_{(G_{ij})}$ **do**

3:     **if** $Q(G'_{ij} \cup G'_{ik}) > Q(G'_{ij})$ **then**

4:       $G''_{ij} = G'_{ij} \cup G'_{ik}$

5:       Remove $G_{ij}$ and $G'_{ik}$ from $R$

6:       Add $G''_{ij}$ to $R_0$

7:     **end if**

8:   **end for**

9: **end for**

10: return $R_0$

---

merged. Then as long as the two subgraphs are merged, the external degree of the input network can be reduced.

## Triangle motif-based clustering procedure

We take the subgraph set of the original network as the input of the motif-based clustering procedure. As shown in Algorithm 3, its main idea is to find a single cluster in a graph by leveraging target motifs. In this procedure, we cluster the subgraphs in $R$ by the minimum conductance, aiming to find the most stable part with the highest conductance of the given subgraph. The algorithm outputs a partition of nodes in $g$ and $\bar{g}$. The motif conductance is symmetric in the sense that $\psi_{\mathrm{Tri}}^{(G)}(G_{\mathrm{Node}}) = \psi_{\mathrm{Tri}}^{(G)}(\bar{G}_{\mathrm{Node}})$, so that any set of nodes ($g$ and $\bar{g}$) can be interpreted as a cluster. However, it is common that one set is substantially smaller than the other in practice. We take the larger set as a module in the network. Some networks are clustered for specific motivations, such as mining the relationships of a person in the social networks. In that case, Algorithm 3 takes the larger part in the clustering results $g$ and $\bar{g}$ as the cluster results as shown from line 9 to line 12.

In the process of the motif-based clustering, we take the target motif and a subgraph partitioned in $R$ (the output of Algorithm 2) as the input. As shown in Fig. 3, for a given graph and a target motif, we calculate a series of matrices, that is, $W_{\mathrm{Tri}}$, $D_{\mathrm{Tri}}$, and $\Gamma_{\mathrm{Tri}}$, before weighting the input graph of matrix $W_{\mathrm{Tri}}$. Therefore, the graph is cut by the minimum conductance $\psi_{\mathrm{Tri}}^{(G)}$ expressed in Eq. (6). In line 8 of Algorithm 3, the value of $\psi_{\mathrm{Tri}}^{(G)}$ is determined by a series of sorted eigenvalues of the subgraph's motif Laplacian matrix $\Gamma_{\mathrm{Tri}}$. Between the two cut subgraphs of the input graph, the larger one will be chosen as the output result.

## Combined COMICS algorithm

In this subsection, we describe the overall algorithm of COMICS in Algorithm 4. It combines all three subprocedures in this subsection.

---

**Algorithm 3** Triangle Motif-based Clustering Algorithm

---

**Input:** Graph $G$ and motif Tri

**Output:** Subgraph set of the original network

1: $(W_{\text{Tri}})_{ij}$ = number of triangle motif instances of Tri

2: $G_{\text{Tri}} \leftarrow$ weighted graph induced $W_{\text{Tri}}$

3: $D_{\text{Tri}}$ = diagonal matrix with $(D_{\text{Tri}})_{ii} = \sum_{j=1}^{n} (W_{\text{Tri}})_{ij}$

4: $\Gamma_{\text{Tri}} = I - D_{\text{Tri}}^{-1/2} W_{\text{Tri}} D_{\text{Tri}}^{-1/2}$

5: $z$ = eigenvector of second smallest eigenvalue for $\Gamma_{\text{Tri}}$

6: $\sigma_i$ = to be the index of $D_{\text{Tri}}^{((-1)/2)}$

7: $z = i$th smallest value

8: $g = argmin_l \ \psi_{\text{Tri}}^{(G)}(G_{Node}^l), \ where \ l = \sigma_1, \cdots, \sigma_k$

9: **if** $|g| > |\bar{g}|$ **then**

10:    return $g$

11: **else**

12:    return $\bar{g}$

13: **end if**

---

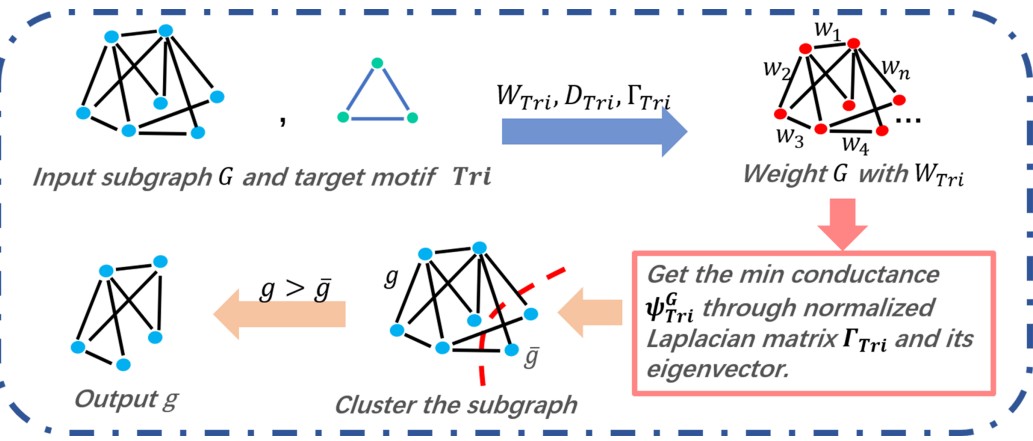

**Figure 3** **Triangle motif-based clustering of COMICS.**

We take the large-scale networks, target motifs and the given validity conditions as the algorithm input, and obtain the clustering set of the original input network. At beginning, a series of partition and refining operations are carried on input networks under the valid conditions. Then we get a partition with high modularity scores of the original input large networks. Each subgraph in the partition set has a strong internal connectivity and a weak external connectivity, maintaining the stable structure information of the original network. Furthermore, we carry the motif-based clustering operations on the subgraph in the partition set. Finally, we can get the non-overlapping optimal partition of the original graph.

## Time complexity analysis

In this section, we analyze the time complexity of COMICS. The main clustering layer includes the following three phases: graph partition, graph refining and motif-clustering.

| Algorithm 4 Combined COMICS Algorithm |
|---|
| **Input:** Large graph $G$, *conditions* and motif Tri |
| **Output:** Motif-based cluster set (subset of nodes in $G$) |
| 1: Set $R_1$ as an empty set |
| 2: $R$ = Graph Partition Algorithm($G$, *conditions*) |
| 3: $R_0$ = Modularity Refine Algorithm($R$) |
| 4: **for** $g$ in $R_0$ **do** |
| 5:    $g'$ = Triangle Motif-based Clustering Algorithm($g$, Tri) |
| 6:    Add $g'$ to $R_1$ |
| 7: **end for** |
| 8: return $R_1$ |

We assume $m$ and $n$ as the number of edges and nodes in the network. Here, $d(v)$ is the degree of node $v$ and $d_{\max}$ is the maximal degree in the network.

**Graph partition procedure:** To get and parse the information from large networks effectively, we first apply cutting operations on the large-scale networks. We analyze the worst case of the graph partition subprocedure. In the first cutting iteration, the root node is one of the nodes with the smallest degree in the graph, and its time complexity is $O(n)$. For each new subgraph $g$ generated by root node $v$, we check the connectivity of the new-added nodes, including the internal and external links with subgraph $g$ and check the corresponding conditions before partitioning. The time complexity of this procedure is $O(n^2 d_{\max}^2)$.

**Modularity refining procedure:** In the subgraph refining procedure, we use modularity score $Q$ (*Newman, 2006*) to get a suitable partition. The required time of iterations is up to the number of subgraphs in the result sets $R$, which are generated by the graph partition procedure. We define the number of subgraphs in $R$ as $p$, and the runtime of the procedure is determined by the step of calculating $Q$, whose computation complexity is $O[(m + n)n]$. Because the first refining subgraph need to check the other $p - 1$ subgraphs and the second one checks the remaining $p - 2$. Hence, the total checking times is $p^2/2$. Therefore, the computation complexity of the refining procedure is $O[p^2(m + n)n/2]$.

**Triangle motif-based clustering procedure:** In general, the time complexity of the algorithm is determined by the construction of the adjacency matrix and the solution of the eigenvector. For simplicity, we consider that we can access network edges within $O(1)$, and modify matrix entries within $O(1)$. The complexity of calculating the eigenvectors through Laplacian matrix is $O((m + n)(\log n)^{O(1)})$, and sorting the eigenvector indexes can be finished in time $O(n \log n)$. For a motif with three nodes, we can compute $W_M$ in $\Theta(n^3)$ in a complete graph with $n$ nodes. Therefore, the computation complexity of the motif-based analysis procedure is $O(n^3)$.

According to the description above, the time complexity of COMICS is $O(rn^3)$, where $r$ is the number of subgraphs in the partition set $R$, and $n$ is the number of nodes.

## EXPERIMENTS

In this section, we compare COMICS with $K$-means and co-authorship team detection algorithm from the perspectives of network clustering accuracy and time complexity,

| | TV shows | Politician | Government | Public figures |
|---|---|---|---|---|
| Node | 3,892 | 5,908 | 7,057 | 11,565 |
| Edge | 17,262 | 41,729 | 89,455 | 67,114 |
| | Athletes | Company | New Sites | Artist |
| Node | 13,866 | 14,113 | 27,917 | 50,515 |
| Edge | 86,858 | 52,310 | 206,259 | 819,306 |

**Table 1 Facebook date sets.**

respectively. We choose four large-scale networks, including two social network, that is, Facebook and gemsec-Deezer data sets (*Leskovec & Krevl, 2014*; *Rozemberczki et al., 2018*) and two academic collaboration networks, that is, APS and MAG data sets.

We analyze the accuracy of the clustering results by calculating compactness and separation. We demonstrate the efficiency of our solution in both academic collaboration and social networks. We also consider other statistical data information of academic networks, TCVs, TPVs, and MSV. All those corresponding metrics are illustrated in this section. All experiments are conducted on a desktop with Intel(R) Xeon(R) CPU E5-2690 v3 @ 2.60 GHz (two processors) and 128 GB memory. The operating system is Window 10, and the codes are written in python.

The American Physical Society data set (2009–2013) consists of 96,908 papers associated with 159,724 scholars in the physical field. Meanwhile, the MAG data set (1980–2015) on computer science includes 207,432 scholars with 84,147 papers in the computer science area. Edges in the academic networks represent two authors have coauthored at least one paper. The Facebook social network data set in our experiments contains eight networks, 134,833 nodes and 1,380,293 edges. We list the eight social networks in Table 1. In that case, we cluster the social networks by the different categories listed in the data set. Gemsec-Deezer data set collected by Deezer (November 2017) is also experimentalized in this paper. This data set contains 143,884 nodes and 846,915 edges from three countries, Romania (41,773 nodes and 125,826 edges), Croatia (54,573 nodes and 498,202 edges) and Hungray (47,538 nodes and 222,887 edges).

## Experiment settings

In this subsection, we describe the settings of our experiments from three aspects, that is, time cost, clustering accuracy and academic teamwork behavior analysis with complete triangle motif in academic areas. In academic collaboration networks, we consider two algorithms. The Facebook social networks do not contain any statistical information. Therefore, we merely compare our method with *K*-means algorithm in the social network:

**K-means clustering algorithm** (*Ding & He, 2004*): This method proves that principal components are the continuous solutions to cluster membership indicators for *K*-means clustering. It takes principal component analysis into the clustering process, which is suitable for the scholar science and social data sets.

**Co-authorship algorithm** (*Reyes-Gonzalez, Gonzalez-Brambila & Veloso, 2016*): This algorithm considers all the principal investigators and collaborators, and defines

**Peer**J Computer Science

[1] *knowledge footprints* of a group are the union of all the backward citations used by group members in all of their papers within a specific time period.

*knowledge footprints*[1] of the groups to calculate the distances between scholars and the group. Based on the distance, the academic groups can be detected in an accurate way. This method iterates all the researchers with their collaborator and institution similarities until they are assigned to a academic team can be applied to understand the self-organizing of research teams and obtain the better assessment of their performances.

To demonstrate the runtime efficiency and the accuracy of our clustering results in large-scale networks, we divide the APS and MAG data sets into different parts with various sizes by years, respectively, so that we can get the collaboration networks with distinct number of nodes (from 1,000 to 200,000). Considering the integrality and veracity of the academic research teams in data sets, we take the whole APS and MAG data sets as the collaboration networks to detect the collaborative relationships.

## Evaluation metrics

To evaluate and analyze the accuracy of network clustering results of our proposed COMICS, we use two metrics, that is, compactness and separation, to evaluate node closeness in clustering results and the distances among clusters. In academic collaboration networks, we combine the statistical paper publishing data with network structures together, and calculate three metrics to find the characteristics discovered through the target triangle motif to uncover the hidden collaboration patterns and teamwork of scholars in academic networks.

**Compactness and separation** (*Halkidi, Batistakis & Vazirgiannis, 2002*) are used to evaluate the accuracy of clustering results by different methods. Compactness is a widely used metric to quantify the tightness of clustering subgraphs, representing the distances between nodes in a subgraph. Separation calculates the distances among the cores of different subgraphs. That is, if a clustering subgraph is with lower compactness value and higher separation value, the subgraph can be detected effectively. Compactness is expressed by Eq. (11),

$$\text{Compactness} = \frac{1}{|R|} \sum_{v_i \in \Omega} |v_i - w|. \tag{11}$$

Here, $R$ is the clustering result set, $v_i$ is one of the nodes in the subgraph, and $w$ is defined as the core of the subgraph cluster, because $w$ is the node with the maximum degree in a cluster. The value of $|v_i - w|$ means the shortest distance between node $v_i$ and the cluster core node $w$. SP is defined as in Eq. (12).

$$\text{Separation} = \frac{2}{k^2 - k} \sum_{i=1}^{k} \sum_{j=i+1}^{k} |w_i - w_j|, \tag{12}$$

wherein, $k$ is the number of subgraphs in the result set and $w_i$ is the core of the given subgraph $i$, which is the same as $w_j$. The value of $|w_i - w_j|$ equals to the shortest distance between $w_i$ and $w_j$.

In collaboration networks, we assume the clusters as academic teams, in which scholars work together. Therefore, three metrics are defined to analyze the collaboration behaviors through triangle motif: TCV, TPV, and MSV.

**TCV**: This metric reflects the tightness and volatility among members in a team. For one scholar $i$ in a team, we define the TCV as follows,

$$\sigma_{co} = \frac{\sum_i^n (co_i - co_{ave})^2}{n}. \tag{13}$$

Herein, $n$ is the number of team members, $co_i$ is the number of scholars that scholar $i$ has collaborated within the same team, and $co_{ave}$ is the average number of collaborators collaborated with scholars in a team.

**TPV**: An academic team with high performance refers that the members in team have published a large number of paper. Similarly, in a stable team, the gaps of published paper numbers among team members are small. To evaluate the academic levels and stability of a team, we define TPV as follows:

$$\sigma_{qtt} = \frac{\sum_i^n (q_i - q_{ave})^2}{n}, \tag{14}$$

where $\sigma_{qtt}$ means scholar $i$'s variance of publishing papers in the detected team, $q_i$ is the number of papers that scholar $i$ has published, and $q_{ave}$ is the average number of papers in the team.

**MSV**: This metric calculates the difference of motif number that the scholar nodes are included in the collaboration networks. We define the MSV as follows,

$$\sigma_{primitive} = \frac{\sum_i^n (t_i - t_{ave})^2}{n}. \tag{15}$$

Herein, $t_i$ is the number of target motif that scholar $i$ owns, and $t_{ave}$ is the average motifs of a team.

To uncover the collaboration patterns mined by triangle motifs among scholars in academic teams, we use the above three arguments to analyze relationships between productions and motifs of the clustered academic teams.

## RESULTS AND DISCUSSION

In this section, we evaluate the experimental results by comparing with $K$-means and co-authorship algorithm in both runtime and the effectiveness. In the view of internal and external connections, we calculate compactness and separation values for each algorithm results.

The time cost results of three networks are shown in Tables 2–4, respectively. "$K$" in the tables represents thousand, for example, "$1K$" means a network with one thousand nodes. N/A means that the clustering procedure takes more than 5 days.

According to Tables 2–4, it can be concluded that, in small networks (less than 30,000 nodes), the three methods make little differences in running time. However, as the size of network increases, our clustering algorithm costs the least time. The time costs in

**Table 2  APS runtime.**

|  | COMICS | Co-authorship | K-means |
|---|---|---|---|
| 1 K | 36.32 s | 2.12 s | 1.73 s |
| 3 K | 435.67 s | 17.45 s | 207.06 s |
| 10 K | 3,058.21 s | 1,084.83 s | 3.47 h |
| 30 K | 1.03 h | 2,856.47 s | 5.73 h |
| 50 K | 1.83 h | 4.82 h | 13.36 h |
| 80 K | 2.29 h | 9.87 h | >24 h |
| 120 K | 5.46 h | 16.36 h | >24 h |
| 150 K | 9.97 h | >24 h | N/A |

**Table 3  MAG runtime.**

|  | COMICS | Co-authorship | K-means |
|---|---|---|---|
| 1 K | 24.74 s | 3.79 s | 2.04 s |
| 3 K | 343.17 s | 21.05 s | 237.93 s |
| 10 K | 2,956.64 s | 345.29 s | 3.53 h |
| 30 K | 1.08 h | 2,636.95 s | 6.62 h |
| 50 K | 2.47 h | 2.93 h | 12.48 h |
| 80 K | 3.35 h | 4.07 h | >24 h |
| 120 K | 5.09 h | 8.27 h | >24 h |
| 150 K | 8.91 h | 21.83 h | N/A |
| 200 K | 14.68 h | >24 h | N/A |

**Table 4  Social network runtime.**

|  | COMICS | K-means |
|---|---|---|
| TV shows | 573.62 s | 322.86 s |
| Politician | 1,394.05 s | 786.42 s |
| Government | 1.03 h | 2.71 h |
| Public figures | 1.26 h | 1.60 h |
| Athletes | 1.58 h | 2.04 h |
| Company | 0.98 h | 3.07 h |
| New sites | 4.78 h | 9.32 h |
| Artist | 6.89 h | 23.42 h |
| Romania | 3.46 h | 17.68 h |
| Hungray | 3.96 h | 18.07 h |
| Croatia | 6.04 h | 38.42 h |

different data sets make little differences. However, the results show the same trend and the proposed method takes more time in small networks and outperforms other large networks. As shown in Tables 2 and 3, when academic collaboration networks contain more than 30,000 nodes, COMICS takes the least time than the other two algorithms. More than that, in social networks, the time cost of our method is also satisfied in large size networks. Therefore, it can be concluded that though the partition operations cost a

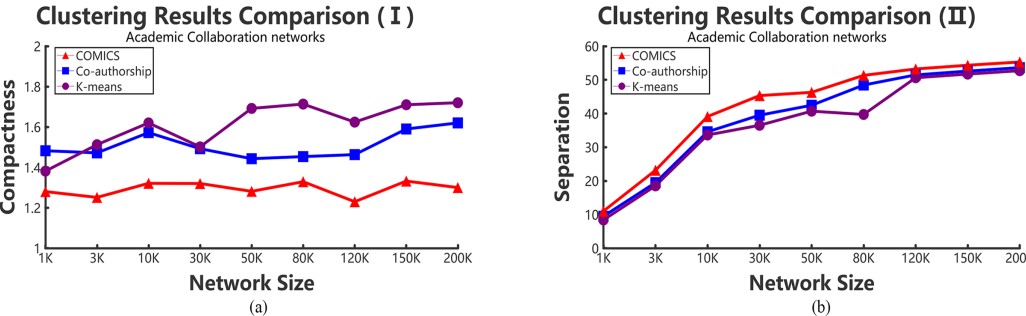

**Figure 4 The variation tendency of compactness and separation values of collaboration network clustering results with COMICS, co-authorship and *K*-means algorithms.** (A) Compactness in academic collaboration networks and (B) separation in academic collaboration networks.

lot of time, it is necessary to apply the speeding up techniques in clustering. Moreover, for different types of networks, topological structures, density are also vital factors that can effect the clustering procedures and results. Figures 4A and 5A show the compactness values generated by our algorithm and the comparing algorithms on different sizes of networks, respectively. As the figures show, in collaboration networks, compactness values corresponding to different networks are lower than those in co-authorship algorithm and *K*-means algorithm, which are similar with that in social networks. Our algorithm performs better than the two comparing algorithms. Figures 4B and 5B plot the separation values of the three algorithms with the network growth in both academic, Facebook social and gemsec-Deezer networks. It can be seen that with the growing network size, COMICS achieves the highest separation values. This means subgraphs clustered by our method have greater separation values all the time. According to Figs. 4B and 5B, we can conclude that the distances among core nodes in each cluster are close no matter what algorithms are used. The reason is that no matter what algorithms are used in the target network, the core nodes of clusters are almost the same. All the core nodes are with the maximum degrees. In all, our clustering algorithm achieves the best subgraph clustering results obviously.

## Analysis in academic collaboration networks

After analyzing the time complexity and effectiveness of our system above, in this subsection, we analyze the clustering results with the triangle motifs in academic collaboration networks. The results prove the triangle motif structures can reflect the hidden statistical information and connections with network structures. For example, as the analysis results show, collaboration patterns as well as the correlations of network structure and team productions can be summarized in the academic collaboration networks.

We regard the cluster results of each academic collaboration network as an academic team. Then the values of three variances, that is, TPV, TCV, and MSV are calculated, and the results are shown in Figs. 6A and 6B. Hence, we can see that the number of high-order triangle motif can reflect the performance of an academic team to some extent.

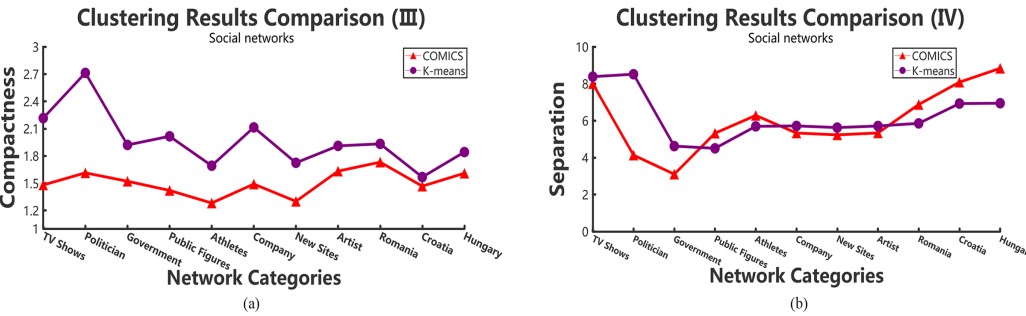

**Figure 5 The variation tendency of compactness and separation values of the clustering results in social networks with COMICS and *K*-means algorithms.** (A) Compactness in social networks and (B) separation in social networks.               

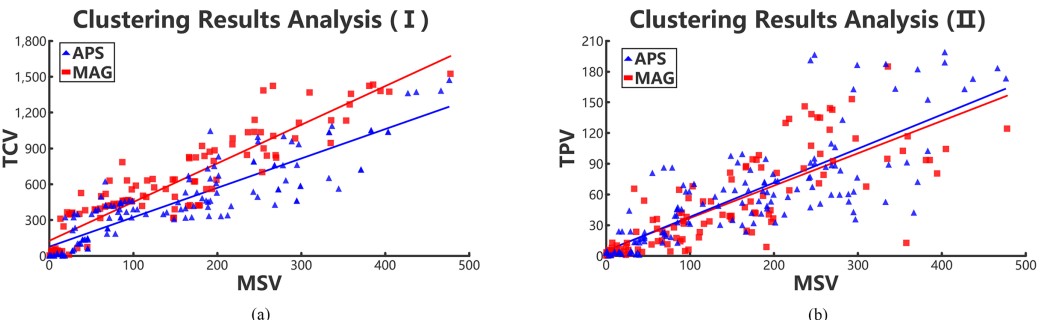

**Figure 6 Positive relations in collaboration networks through collaboration variances, paper variances and motif variances of each clustering.** Red rectangles and blue triangles represent the collaboration academic teams clustered from MAG and APS data sets, respectively. (A) Relationships between TCV and MSV and (B) relationships between TPV and MSV.

According to Figs. 6A and 6B, we conclude that the TPV and TCV are both proportional to the MSV. Meanwhile, the TPV is also approximately positive linear with the MSV. That means, the lower the MSV is in a cluster team, the performance of team members are in smaller gaps. Therefore, it can be concluded that the value of MSV can reflect the gap of collaboration relationships in teams and performance of team members. However, we can infer that the scholars with few number of complete triangle motifs, have collaborated with only few scholars in the team. Those scholars are probably students or new team members, resulting in the high collaboration and paper variances. Hence, in collaboration networks, we can use MSV to evaluate the gaps of team collaboration relationships and the performance of team members. The two teamwork gaps in different periods represent the stability and volatility of academic teams.

## CONCLUSION

In this paper, we put forth the high-order motif-based clustering system to get a subgraph set from the large-scale networks. In the constructed system, we take graph partition and refining techniques to speed up algorithm runtime. Through network cutting, we check the

four cutting conditions from the aspect of network connectivity, which can prevent damaging the global structures of large-scale networks. Experiments are carried on four large networks, that is, APS and MAG from the academic area, Facebook and gecsec-Deezer networks from the social area, respectively. The results demonstrate the effectiveness of our method in time cost and accuracy in large-scale network clustering.

Furthermore, the collaboration teamwork analysis verifies the availability of complete triangle motif, which represents the smallest collaboration unit in the collaboration networks. We analyze the collaboration clustering results with three metrics, that is, TCV, TPV, and MSV. The results show that both TCV and TPV are proportional to MSV. Therefore, it can be concluded that the value of MSV can reflect the two gaps, that is, collaborative relationships and performance of different team members. Besides, the two gaps in different periods can also reflect the dynamic change of team members. In the future, we will focus on dynamic motif clustering for real-time network management (*Ning et al., 2018*; *Ning, Huang & Wang, 2019*; *Wang et al., 2018a*). In addition, network security (*Wang et al., 2018b*, *2019*) and crowdsourcing based methods (*Ning et al., 2019a*, *2019b*) also deserve to be investigated.

### Funding

This work is supported by China Postdoctoral Science Foundation under Grant 2018T110210 and State Key Laboratory for Novel Software Technology, Nanjing University, under Grant KFKT2018B04. The funders had no role in study design, data collection and analysis, decision to publish, or preparation of the manuscript.

### Grant Disclosures

The following grant information was disclosed by the authors:
China Postdoctoral Science Foundation: 2018T110210.
State Key Laboratory for Novel Software Technology, Nanjing University: KFKT2018B04.

### Competing Interests

The authors declare that they have no competing interests.

### Author Contributions

- Yufan Feng conceived and designed the experiments, performed the experiments, analyzed the data, contributed reagents/materials/analysis tools, prepared figures and/or tables, authored or reviewed drafts of the paper, approved the final draft.
- Shuo Yu conceived and designed the experiments.
- Kaiyuan Zhang analyzed the data, contributed reagents/materials/analysis tools, performed the computation work.
- Xiangli Li performed the experiments, analyzed the data, performed the computation work.
- Zhaolong Ning conceived and designed the experiments, authored or reviewed drafts of the paper, approved the final draft.

## Data Availability

Data is available through the American Physical Society (APS) and Microsoft Academic Graph (MAG). Specifically:

Facebook network data can be found here: http://snap.stanford.edu/data/ego-Facebook.html.

Gemsec-Deezer network data can be found here: http://snap.stanford.edu/data/gemsec-Deezer.html.

Code can be found here:

https://github.com/yffre/COMICS.

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
