# Peer review of "COMICS: a community property-based triangle motif clustering scheme"

_PeerJ Computer Science, doi:10.7717/peerj-cs.180_

## Round 0.1 · original submission · Major Revisions

Two reviews have been obtained. Both reviewers thought this paper has some merits but also mentioned miscellaneous issues in presentation as well as technical aspects. Besides, it is recently reported in the work Clustering coefficients of large networks (Information Sciences) that many real-world large scale networks have a common clustering feature reflected in their clustering coefficients. Do the networks considered in the current work belong to this category, and why? How does this potentially affect the proposed triangle motif clustering scheme?

Reviewer 1 ·

Basic reporting

--The outline of the article is clear in general. However, there are quite a few issues in the presentation that make careful readers fail to understand the content thoroughly. I list some while there are a lot left:
*In the intro of the roadmap of this article, all the reference for section numbers are missing. It is the same in other places all throughout the article.
*In Figure 2, the subfigures in the second line are exactly the same except for some literal explanations. The subfigures themselves have no usage.
*In Algorithm 1, what is the definition of G'? Is this G' same as that the problem statement? It is also hard to understand the inner loop of Algorithm 1 and its corresponding explanation in lines from 197-200.
* What is the definition of R0 in algorithm 2? R is changing due to the steps 4, 5. Is R0 changing as well? If R0 does not change, does it mean that it is at most two subgraphs that may merge? If so, it does not seem to be rational?
* The complexity analysis is kind of a mess. For Alg1, is searching for the smallest degree O(1)? For Alg2, again, the authors seem to assume that at most two subgraphs may merge. For Alg 3, if only the second smallest eigenvalue is computed, the complexity is just O(n+m), and Benson's Science paper shows that there will be more efficient methods to search for triangles (better than O(n^3)).
* The baseline algorithms including K-means and co-authorship algorithms are explained unclearly. For example, what are the features used in K-means...
* The x-axis in Figure 5 is incorrect.
* Line 324, why w (the core) is the node with the maximal degree?

-- References: some important references are missing.
1. Investigation of triangle motifs in clustering over social networks was also discussed in quite a few literatures such as
"Motif Clustering and Overlapping Clustering for Social Network Analysis" by Pan Li et al. (INFOCOM17)
2. Conductance for motifs is essentially conductance for hypergraphs. Related important works are in
"Hypergraph Markov Operators, Eigenvalues and Approximation Algorithms" by Anand Louis (FOCS15)
"Submodular Hypergraphs: p-Laplacians, Cheeger Inequalities and Spectral Clustering" by Pan Li et al. (ICML18)
3. Alg3 on motif-based clustering essentially comes from learning on hypergraphs.
"Learning with Hypergraphs: Clustering, Classification, and Embedding" by Zhou et al. (NIPS07)

--Other typos:
equation 3: G_t -> G_k
line 188: Algorithm in ?
line 379: Fig.6b -> Fig 6a and Fig 6b
line 381: the smaller gap of team member performance? Grammarly incorrect!

Experimental design

This work is to propose a clustering algorithm for social networks and co-authorship networks and investigate more on the function of triangle motifs in social networks and co-authorship networks.

Compared to Benson's Science work, the algorithmic contribution is incremental. Some tricks are used in the clustering algorithms (Alg1+Alg2), while the authors did not explain well why they work. I suggest that the authors provide more explanations from both mathematical and experimental perspectives on these tricks. The baselines used for comparing the clustering performance are poor. I suggest that the authors compare the results based on sole Alg1, Alg1+Alg2, Alg1+Alg2+Alg3, sole Alg3 to let readers understand what is happening.

The good thing about this work is to provide some new comparison that may help researchers understand the function of triangle motifs. I suggest the authors provide more comparison from this point. I also suggest the authors provide additional experiments on other social network datasets (more than FB).

Validity of the findings

I think the impact and novelty are incremental unless the authors may provide better explanations of their clustering algorithms, especially the necessity of Alg1 and Alg2, and the authors may provide more evidence on the functions of triangle motifs.

Reviewer 2 ·

Basic reporting

The paper is interesting and has novelty. It is well structured and well written in general. However it needs some extensions improvements and clarifications.

I suggest authors explicitly highlight their motivation to conduct the present work and also what is the exact motivation of the work.

How do the findings of the study contribute to the body of knowledge? Please present in better detail the functionality of the property-based triangle motif clustering scheme and how it would be assistive to the related research community.

The literature needs a better reviewing since there are many related quality works that authors could examine. Please examine additional works on the community detection which utilize also user characteristics.

Please point out the novelty of the work.

The results need a better discussion and the impact of the work needs to be highlighted.

Experimental design

In the experimental study please substantiated the choose of the three large scale networks that were utilized.

The results need a deeper discussion.

What are the reasons behind the performance of authors approach? Please explain in detail the performance results presented in figures 5&6.

Validity of the findings

The findings are interesting. However, the results need a better and deeper discussion.

Additional comments

The paper is interesting and has novelty. It is well structured and well written in general. However it needs some extensions improvements and clarifications.

I suggest authors explicitly highlight their motivation to conduct the present work and also what is the exact motivation of the work.

How do the findings of the study contribute to the body of knowledge? Please present in better detail the functionality of the property-based triangle motif clustering scheme and how it would be assistive to the related research community.

The literature needs a better reviewing since there are many related quality works that authors could examine. Please examine additional works on the community detection which utilize also user characteristics.

Please point out the novelty of the work.

The results need a better discussion and the impact of the work needs to be highlighted.

In the experimental study please substantiated the choose of the three large scale networks that were utilized.

The results need a deeper discussion.

What are the reasons behind the performance of authors approach? Please explain in detail the performance results presented in figures 5&6.

The findings are interesting. However, the results need a better and deeper discussion.

---

## Round 0.2 · Minor Revisions

One reviewer has no further comments. The other reviewer suggests some further modifications. Besides, the authors overlooked my comments in the revised version. Please consider to address in the re-revised version. It is recently reported in the work Clustering coefficients of large networks (Information Sciences) that many real-world large scale networks have a common clustering feature reflected in their clustering coefficients. Do the networks considered in the current work belong to this category, and why? How does this potentially affect the proposed triangle motif clustering scheme?

Reviewer 1 ·

Basic reporting

The manuscript is improved a lot after authors' first-round revision overall. Some minor issues are left and list as the follows.
* For the added references to hypergraphs, such as Zhou et al. NIPS07, I suggest the authors may explain well the relation that graphs + motif-based structures essentially give hypergraphs. To understand this point, the authors can check Section S1.12 in the supplementary material of Benson's work in Science cited in this paper or the Introduction of "inhomogeneous hypergraph clustering with applications" by Pan Li et al. NIPS'17.
* Tri_2 in Figure 1: The set A should be {1, 3} according to the format of B_2 which shows node 2 is connected with node 1 and node 3. Please make them consistent.
* The authors should also revise their conclusion accordingly as they add new datasets. Please go through the manuscript to make all statements consistent with the new revisions.

For typos, the authors should carefully proofread the manuscript several times to make sure there are not typos. I found a lot even in the new added parts:
* line 195, "there is only graph are added" is not corrected in grammar.
* Algorithm 1, G_ij -> G_ij'
* line 222, do es -> does

Experimental design

The added explanation of Alg 1 and Alg 2 is greatly helpful. I feel pleased to see that.

Validity of the findings

As the explanations of Alg 1 and Alg 2 give insight, now I think the manuscript has novelty and impact.

Reviewer 2 ·

Basic reporting

no comment

Experimental design

no comment

Validity of the findings

no comment

Additional comments

Authors have addressed my previous comments.

---

## Round 0.3 · accepted · Accept

I checked the revised paper and the response letter of the authors. I am confident that all the comments have been addressed.